# An Examination of Serum Acylcarnitine and Amino Acid Profiles at Different Time Point of Ketogenic Diet Therapy and Their Association of Ketogenic Diet Effectiveness

**DOI:** 10.3390/nu13010021

**Published:** 2020-12-23

**Authors:** Pi-Lien Hung, Ju-Li Lin, Chien Chen, Kai-Yin Hung, Tzu-Yun Hsieh, Mei-Hsin Hsu, Hsuan-Chang Kuo, Ying-Jui Lin

**Affiliations:** 1Department of Pediatrics, Division of Pediatric Neurology, Kaohsiung Chang Gung Memorial Hospital, Chang Gung University College of Medicine, Kaohsiung 833, Taiwan; flora1402@cgmh.org.tw (P.-L.H.); a03peggy@cgmh.org.tw (M.-H.H.); 2Department of Pediatrics, Division of Genetics and Endocrinology, Linkou Chang Gung Memorial Hospital, Chang Gung University College of Medicine, Taoyan 333, Taiwan; lin001@cgmh.org.tw; 3Department of Neurology, Taipei Veterans General Hospital, National Yang-Ming University, Taipei 112, Taiwan; cchien0604@gmail.com; 4Division of Nutritional Therapy, Kaohsiung Chang Gung Memorial Hospital, Kaohsiung 833, Taiwan; redrosahung@cgmh.org.tw; 5Department of Pediatrics, Division of Pediatric Critical Care, Kaohsiung Chang Gung Memorial Hospital, Chang Gung University College of Medicine, Kaohsiung 833, Taiwan; role0302@cgmh.org.tw (T.-Y.H.); kuohc117@cgmh.org.tw (H.-C.K.)

**Keywords:** ketogenic diet, free carnitine, acylcarnitines, amino acids, mitochondrial βoxidation function

## Abstract

Background: This study aimed to identify metabolic parameters at different time points of ketogenic diet therapy (KDT) and investigate their association with response to KDT in pediatric drug-resistant epilepsy (DRE). Methods: Prospectively, twenty-nine patients (0.67~20 years old) with DRE received classic ketogenic diet with non-fasting, gradual KD initiation protocol (GRAD-KD) for 1 year were enrolled. A total of 22 patients remaining in study received blood examinations at baseline, 3rd, 6th, 9th, and 12th months of KDT. β-hydroxybutyrate, free carnitine, acylcarnitines, and amino acids were compared between responders (seizure reduction rate ≥ 50%) and non-responders (seizure reduction rate < 50%) to identify the effectiveness of KDT. Results: The 12-month retention rate was 76%. The responders after 12 months of KDT were 59% (13/22). The free carnitine level decreased significantly at 9th months (*p* < 0.001) but increased toward baseline without symptoms. Propionyl carnitine (C3), Isovaleryl carnitine (C5), 3-Hydroxyisovalerylcarnitine (C5:OH) and methylmalonyl carnitine (C4-DC) decreased but 3-hydroxybutyrylcarnitine (C4:OH) increased significantly at 12th months of KDT. The glycine level was persistently higher than baseline after KDT. KDT responders had lower baseline C3 and long-chain acylcarnitines, C14 and C18, as well as lower C5, C18, and leucine/isoleucine. Conclusions: KDT should be avoided in patients with non-ketotic hyperglycemia. Routine carnitine supplementation is not recommended because hypocarnitinemia was transient and asymptomatic during KDT. Better mitochondrial βoxidation function associates with greater KDT response.

## 1. Introduction

Drug-resistant epilepsy (DRE) refers to epilepsy that does not respond to two or more antiepileptic drugs and accounts for 20–30% of childhood epilepsy. Thirty-three percent of patients with DRE who receive classic ketogenic diet therapy (KDT) have a more than 50% reduction in seizures and 15–20% become seizure-free [1,2,3,4,5]. KD is also thought to be effective in two metabolic disorders, i.e., glucose transporter type 1 deficiency syndrome, pyruvate dehydrogenase deficiency and in certain mitochondrial disorders [6]. However, with exception of the above metabolic and brain disorders, the response to KDT cannot be predicted.

Specific electroencephalography parameters had been reported to predict a favorable response to KDT in patients with DRE [7]. Biochemical parameters, including glucose and plasma β-hydroxybutyrate (βHB) levels, the ratio of blood glucose to ketones, and the glucose ketone index had been reported to be associated with the efficacy of KDT in brain cancer in vivo [8]. Schoeler et al. reported a positive association between a higher baseline acetyl-carnitine level and greater efficacy of KDT [9]. However, there is limited evidence for an association between biochemical parameters and the effectiveness of KDT. Prolong KDT produces ketosis with increasing formation of acylcarnitines from free carnitine, leading to a drop in plasma free carnitine [10]. However, there were also scanty evidences regarding association between free carnitine and KD response.

There are several possible biochemical explanations for how KDT could control seizures. It had been reported that the antiepileptic effect of KDT results from a shift in the biogenesis of the mitochondria to a less hyperexcitable state. KDT restricts sugar and produces ketones that interact with receptors, channels, and enzymes [11,12,13]. Changes in neurotransmitter and amino acid levels in cerebrospinal fluid (CSF) in response to KDT may have anticonvulsant effects [14]. One report suggested that changes in the brain glutamate level brought about by KDT help to diminish seizures [15,16]. Furthermore, differences in CSF lysine and arginine levels had been found between responders and non-responders to KDT for epilepsy. Amino acids seem to play an important role in the antiepileptic effect of KDT; however, unlike for CSF, there are few reports on the changes in blood amino acid levels, in response to KDT [14]. Jirapinyo et al. found a high serum branched-chain amino acid/aromatic amino acid ratio after 10 days of KDT in a pediatric population [17]; however, the association between serum amino acids and the effectiveness of KDT is still under investigation. We hypothesized that some metabolic parameters at baseline or during KDT can predict or be associated with response to KDT. The aim of this study was to identify predictors and factors associated with the effectiveness of KDT in DRE.

## 2. Materials and Methods

### 2.1. Participants

Children with DRE were enrolled in this single-center, prospective, cohort study to receive a ketogenic diet from January 2018 to January 2019. The study was supervised by the Committee of Institutional Review Board at Chang Gung Memorial Hospital (IRB no.201700968A3). Informed consent was obtained from all guardians on enrollment.

DRE was defined as failure of adequate trials of two tolerated and appropriately used antiepileptic drug (AED) schedules (either as monotherapy or in combination) to achieve sustained seizure freedom [18]. Children with a confirmed diagnosis of inborn error of metabolism (pyruvate carboxylase deficiency, primary carnitine deficiency, fatty acid oxidation deficiency, ketolysis deficiency, familial hypercholesterolemia) and those who were surgical candidates were excluded.

### 2.2. Study Protocol

All enrolled patients were admitted to a pediatric general ward at Kaohsiung Chang Gung Memorial Hospital to receive a 5-day diet program. All study subjects underwent metabolic screening, including blood ammonia and lactate levels, serum cholesterol and triglyceride levels, urinary organic acids, and blood spot tests for amino acid profiles to exclude medical conditions that were inappropriate for KDT. All subjects were screened for a family history of porphyria during the initial visit. Classic ketogenic diet (cKD) was initiated in each patient with GRAD-KD protocol during the 5-day hospitalization. A registered dietitian calculated the energy requirement based on daily activity, dietary history, age, body weight and height. During the 5-day admission, patients received one-ninth of Recommended Dietary Allowance (RDA) calories as KD prescription on day 1, one-sixth of RDA calories on day 2, one-third of RDA calories on day 3, two-thirds of RDA calories on day 4, and the full RDA calories on day 5. The keto ratio started at 2:1 on day 1 and gradually increased to 3~4: 1 on day 5, which approached 85% fat, 12% protein and 3% carbohydrate. The daily protein intake, predominantly animal protein, was in the range of 1.5–2.5 g/kg/day. Medium chain triglyceride powder was prescribed gradually at a minimum dosage of 40 g/day if patients can tolerate it. Blood sugar and βHB levels were monitored every 2–4 h during the hospitalization period to avoid hypoglycemia and hyperketosis. The blood βHB level was obtained from a fingertip blood sample (Free Style Optium Neo Blood Glucose and Ketone Monitoring System; Abbott Diabetes Care Inc., Whitney, UK) taken on the 2nd day of the 5-day diet program. On the day of discharge, the KDT nursery team educated the families on how to measure the urinary ketone level using a dipstick (Ketostix^®^, Bayer Diabetes, Berkshire, UK). Seizure frequency and βHB level were recorded by seizure diaries at monthly follow-up visits for 12 months.

### 2.3. Determination of Response to KDT

The response to KDT was estimated in 28-day epochs for 12 months. Monthly seizure frequencies were calculated from seizure diaries. The formula used to calculate seizure reduction rate was [(a–b)/a] × 100 where “a” is the number of seizures in the 28-day baseline period and “b” is the number of seizures in the 28 days before the 12-month follow-up visit [9]. Patients with a reduction in seizure frequency of ≥50% were defined as responders, and those with a reduction of <50% were defined as non-responders.

### 2.4. Tandem Mass Spectrometry

Blood samples were taken after 10 h of fasting at baseline and at 3, 6, 9, and 12 months for measurement of fatty acids and amino acids by tandem mass spectrometry. The samples were prepared by removing the entire dried blood sample using a standard 0.3-cm single-hole punch into a 1-mL 96-well plate (Waters Corp., Milford, MA, USA; catalog number WAT058957). A 200-µL working standard (1:490 diluted in methanol; Cambridge Isotope Laboratories, Tewksbury, MA, USA; catalog numbers NSK-A and NSK-B) was placed in the 96-well plate. The plate incubated on a microplate shaker (Taitec, Tokyo, Japan) at a speed of 400–450 rpm. The supernatant was transferred to a Nunc plate (catalog number 249,944; Nunc, Roskilde, Denmark) and dried using a TurboVap (Caliper life Sciences, Hopkinton, MA, USA) at 70 °C. Samples were reconstituted using 60 μL of 80% deionized water in acetonitrile, mixed using the Wellmix on high speed for 1 min. The mass analysis was performed using a Micromass Quattro Ultima Pt mass spectrometer (Waters 2795). Analyte-specific mass spectrometry parameters were optimized for the highest signal intensity. The data were analyzed using Masslynx 4.0 version software (Waters).

### 2.5. Statistical Analysis

The data are presented as the mean ± standard error of the mean. Demographic parameters were examined by Fisher’s exact test. Biochemical parameters, including βHB and plasma acylcarnitines [19] and amino acid levels were compared in and between groups at the various time points by conducting generalized estimating equation (GEE) after adjusting confounding effect of anticonvulsants. Associations between βHB levels, free carnitine levels, and seizure outcomes were analyzed by the Spearman zero order correlation coefficient. Receiver-operating characteristic (ROC) curve analysis was used to identify metabolites that are associated with the effectiveness of KDT. The area under the curve (AUC) values were determined using the receiver-operating characteristic ROC curve. The 95% confidence interval of area under curve (AUC) was produced by using the pROC [19] package in R (version 3.5.2; R Project for Statistical Computing, https://www.r-project.org). All statistical analysis was performed using SPSS for Windows version 19.0 software (IBM Corp.). *p* < 0.05 was considered statistically significant.

## 3. Results

### 3.1. Clinical Characteristics

Seven of the 29 patients who met the diagnostic criteria for DRE did not complete the 12-month study. These seven patients withdrew before completion of 9 months either due to diet intolerance or parent-related or child-related reasons. Twenty-two patients completed 12 months of KDT. The diet retention rate at 3, 6, 9, and 12 months were 89.6%, 83%, 76%, and 76%, respectively (Figure 1). The clinical characteristics of the 22 patients were shown in Appendix A. The mean patient age was 9.3 ± 1.98 years (range, 8 months to 19 years), and the male to female ratio was 1.8:1. Generalized seizures occurred in 36.3% (8/22) of patients, focal seizures in 41% (9/22), and infantile spasm in 23% (5/22). The leading etiology was genetic (31.8%, 7/22) followed by hypoxic-ischemic encephalopathy (22.7%, 5/22). Five (22.7%, 5/22) of patients had a seizure reduction rate of ≤25%, four (22.7%, 4/22) had a reduction of 25–50%, nine had a reduction of 50%–90%, and four (18.2%, 4/22) had a reduction of ≥90%. The response rate (seizure reduction rate ≥50%) to 12 months of KDT was 59% (13/22), with three patients (13.6%, 3/22) becoming seizure-free.

### 3.2. Metabolic Changes after KDT

All biochemical parameters were adjusted by GEE to exclude confounding effect of anticonvulsants (AEDs). Plasma acylcarnitines, and amino acids level at different time point of KDT were compared with those at baseline (Table 1).

### 3.3. Short-Chain Acylcarnitines after KDT

Acetylcarnitine (C2), an important substrate for mitochondrial energy metabolism, was significantly higher than baseline level from 3 months to 12 months (C2, β = 7.14, *p* < 0.05). Propionate acyl-carnitine (C3) and methylmalonyl carnitine (C4-DC) levels were significantly lower than baseline value from 3 months to 12 months (C3, β = −0.79, *p* < 0.001; C4-DC, β = −0.31, *p* < 0.001). 3-hydroxybutyrylcarnitine (C4:OH), derives from CoA-ester of the ketone body, D-3-hydroxybutyrate [20], was significantly higher than baseline value from 3 months to 12 months (C4:OH,β = 0.24, *p* < 0.001). Isovaleryl carnitine (C5) was significantly lower than baseline value from 9 months to 12 months (C5, β = −0.02, *p* < 0.05). 3-Hydroxyisovalerylcarnitine (C5:OH) was significantly lower than baseline value from 6 months to 12 months (C5:OH, β = −0.04, *p* < 0.05). C3, C5, and C5:OH are degradation products of ketogenic amino acids: lysine, tryptophan, leucine and isoleucine [21].C4-DC is derived from branch-chain amino acids metabolism: leucine, isoleucine and valine which are ketogenic or glucogenic amino acids, showed a positive correlation of basal glucose level [22]. The results herein disclosed that KDT successfully led to ketogenesis by increasing CoA-ester formation from fatty acid β oxidation instead of degrading ketogenic amino acids. On the other hand, increasing acetylcarnitine production in our results can explain how KDT could alter mitochondrial energy metabolism.

### 3.4. Medium-Chain Acylcarnitines after KDT

Octanoyl-(C8)/decanoyl-(C10) acyl-carnitine levels were significantly lower than baseline values at 12 months (C8/C10, β = −0.15, *p* < 0.005). We supposed that the C8/C10 ratio became lower after KDT may be due to highest net ketogenic effect of C8 over another medium-chain triglyceride [23] so that C8 was rapidly turn over into ketone body.

### 3.5. Long-Chain Acylcarnitnes after KDT

C14 carnitine was significantly lower than baseline values at 6 months and at 9 months; however, it became not significantly different from baseline value at 12 months (C14, β = −0.03, *p* < 0.005 at 6 mons; β = −0.02, *p* < 0.05 at 9 months). C14:1 was transiently higher than baseline values at 3 months (C14:1, β = 0.03, *p* < 0.01). C14:2 carnitine was significantly higher than baseline values at 12 months (C14:2, β = 0.01, *p* < 0.05). C16 carnitine was significantly lower than baseline values from 6 months to 12 months (C16, β = −0.26, *p* < 0.005 at 6 months; β = −0.31, *p* < 0.01 at 9 months; β = −0.38, *p* < 0.01 at 12 months). C16:1 carnitine was significantly lower than baseline values at 6 months and 12 months (C16:1, β = −0.02, *p* < 0.05 at 6 months; β = −0.03, *p* < 0.05 at 12 months). C18 carnitine showed no significant changes after KDT. Our data demonstrated that C14:1 and C14:2 carnitine levels trended upwards during the study period.

### 3.6. Amino Acids after KDT

After 12 months of KDT, there was a significant decrease in levels of ketogenic amino acids, including phenylalanine, tyrosine, and leucine/isoleucine (Phenylalanine, β = −6.63, *p* < 0.001; tyrosine, β = −13.79, *p* < 0.001; leucine/isoleucine, β = −48.35, *p* < 0.001). Methionine, a substrate for biogenesis of carnitine, was significantly lower than baseline value from 3 months to 12 months of KDT (Methionine, β = −5.48, *p* < 0.001). It shared the similar tendency of free carnitine in our study. Glycine was the unique amino acid which increased its level significantly with compared to baseline levels from starting KD to 12 months of KDT (Glycine, β = 78.21, *p* < 0.001 at baseline; β = 91.01, *p* < 0.05 at 3 months; β = 74.71, *p* < 0.005 at 6 months; β = 49.82, *p* < 0.01 at 12 months). The glucogenic amino acid, valine, showed significantly lower than baseline values at 9 months and 12 months of KDT (valine, β = −38.38, *p* < 0.05 at 9 months; β = −41.06, *p* < 0.05 at 12 months). In addition, proline was transiently lower than baseline at 6 months and 9 months (proline, β = −52.66, *p* < 0.001 at 6 months; β = −49.16, *p* < 0.005 at 12 months), but it increased at 12 months and showed no significant differences compared to baseline values. Arginine was transiently lower at 6 months, whereas citrulline and homocysteine were transiently higher at 3 months (arginine, β = −0.90, *p* < 0.05 at 3 months; citrulline, β = 2.57, *p* < 0.05 at 3 months; homocysteine, β = 1.17, *p* < 0.05 at 3 months). However, the serum level of these three amino acids showed no significant changes compared to baseline values at 12 months of KDT. Our results demonstrated that the concentrations of ketogenic amino acids and some glucogenic amino acids became significantly lower after 12 months of KDT. In addition, glycine was found persistently higher than baseline throughout the KDT period.

### 3.7. Free Carnitine after KDT

There was a significant decrease in the free carnitine level after 9 months of KDT, and never returned to baseline level at the end of study (free carnitine, β = −5.10, *p* < 0.01 at 9 months; β = −4.07, *p* < 0.05 at 12 months). However, none of our study subjects had symptoms of hypocarnitinemia, such as muscle weakness, fatigue, or anorexia. None of our study subjects received carnitine supplement during the KDT.

### 3.8. Differences in Metabolic Parameters between Responders and Non-Responders

Demographic variables and clinical characteristics were compared between responders and non-responders at the different time points in Appendix A. There was no significant between-group difference in type of seizure, etiology of epilepsy, body weight, and numbers of AEDs administered. Non-responders received more valproic acid (VPA) as monotherapy or adjuvant therapy than responders did (6 vs. 2, *p* < 0.05).

We offered a comparison of acylcarnitines and amino acids level between responders and non-responders as shown in Table 2. All the comparisons were examined by conducting GEE after adjusting the confounding effect of anticonvulsants. The results were demonstrated in Table 2 and were summarized as following paragraphs.

### 3.9. Free Carnitine and βHB Levels in Responders

There was a significant increase in the βHB level in responders at 6 months (βHB, β = 1.43, *p* < 0.05 at 6 months). Free carnitine level in responders was significantly lower than that in non-responders after initiating of KDT (Free carnitine, β = −7.85, *p* < 0.05 at baseline; β = −11.84, *p* < 0.005 at 3 months; β = −15.94, *p* < 0.001 at 9 months; β = −6.77, *p* < 0.05 at 9 months; β = −8.59, *p* < 0.01 at 12 months). Free carnitine levels reached a nadir in responders at 6 months (Appendix A, responders vs. non-responders, 20.84 ± 6.89 vs. 29.03 ± 8.86, *p* < 0.05), approaching the diagnostic criteria of carnitine deficiency defined as carnitine level ≤20 µM or a ratio of acylated-to-free carnitine of ≥0.4, at an age over 1 week post term [24], and gradually increased its level thereafter. There was a negative correlation between βHB and free carnitine levels (Appendix A, ρ = −0.686, *p* < 0.05). Carnitine is a crucial factor in the transfer of fatty acids for mitochondrial β-oxidation, and βHB is the major product of ketosis, we examined the βHB and free carnitine levels at 6 months to assess the association with efficacy of diet (≥50% seizure reduction). However, neither free carnitine nor βHB at 6 months had clear association with the efficacy of KDT (Appendix A, correlation coefficient of free carnitine and seizure reduction rate, ρ = −0.109, *p* = 0.781; correlation coefficient of βHB and seizure reduction rate, ρ = −0.427, *p* = 0.252).

### 3.10. Short-Chain Acylcarnitines in Responders

There was a decrease in C2 in responders at baseline and at 3 months, however the C2 levels became comparable between groups after 3 months (C2, β = −8.95, *p* < 0.005 at baseline; β = −9.67, *p* < 0.05). C3 levels in responders were significantly lower after initiating of KDT to the end of study (C3, β = −1.28, *p* < 0.05 at baseline; β = −1.68, *p* < 0.001 at 3 months; β = −1.8, *p* < 0.001 at 6 months; β = −1.12, *p* < 0.05 at 9 months; β = −0.98, *p* < 0.05 at 12 months). C4:OH levels were significantly lower in responders than in non-responders at baseline, but the level became comparable between groups after initiating of KDT (C4:OH, β = −0.07, *p* < 0.05). C4-DC levels in responders were transiently and significantly lower at 3 months, and the levels became comparable between groups from 3 months to 12 month (C4-DC, β = −0.06, *p* < 0.05 at 3 months). C5 levels in responders were significantly lower than that in non-responders from 3 to 12 months (C5, β = −0.09, *p* < 0.001 at 3 months; β = −0.10, *p* < 0.001 at 6 months; β = −0.07, *p* < 0.005 at 9 months; β = −0.04, *p* < 0.005 at 12 months). C5:1 level was fluctuating in responders. C5:1 levels were significantly lower in responders at 3 months and 9 months, but they showed no significant differences between groups at baseline, 6 months, and 12 months (C5:1, β = −0.01, *p* < 0.05 at 3 months; β = −0.01, *p* < 0.05 at 9 months). C5-DC levels were significantly lower in responders at 3 months, 9 months, and 12 months (C5-DC, β = −0.09, *p* < 0.001 at 3 months; β = −0.02, *p* < 0.05 at 9 months; β = −0.01, *p* < 0.01 at 12 months). Our data demonstrated that C3 and C5 carnitines had persistently lower levels in responders than in non-responders after initiating of KDT.

### 3.11. Medium-Chain Acylcarnitines in Responders

We found C6 levels were significantly lower in responders at 6 months and 9 months, but there were no between-group differences at other time point (C6, β = −0.02, *p* < 0.001 at 6 months; β = −0.02, *p* < 0.05 at 9 months). C6-DC levels were significantly lower in responders only at 6 months, but there were no significant between-group differences at other time point (C6-DC, β = −0.01, *p* < 0.05 at 6 months). C8, C10, and C10:1 levels were significantly lower in responders at 9 months, but there were no significant between-group differences at other time point (β and *p* value at 9 months were shown as follows: C8, β = −0.06, *p* < 0.001; C10, β = −0.07, *p* < 0.001; C10:1, β = −0.08, *p* < 0.001). C12 levels were significantly lower in responders only at 3 months, but not at other time points (C12, β = −0.07, *p* < 0.05). In brief summary, although medium-chain acylcarnitines showed significantly lower in responders at some time points of KDT; however, none of them showed significant differences between responders and non-responders at endpoint of the study.

### 3.12. Long-Chain Acylcarnitines in Responders

C14 levels in responders were also significantly lower than that in non-responders at baseline, 6 months and 9 months (C14, β = −0.04, *p* < 0.05 at baseline; β = −0.04, *p* < 0.01 at 6 months; β = −0.04, *p* < 0.05 at 9 months). C14:1levels were significantly lower in responders at baseline, 3 months and 9 months (C14:1, β = −0.04, *p* < 0.001 at baseline; β = −0.05, *p* < 0.05 at 3 months; β = −0.05, *p* < 0.01at 9 months). C14:2 levels were significantly lower in responders at 9 months, but there were no between-group significant differences at other time points (β = −0.03, *p* < 0.005 at 9 months). However, both C14:1 and C14:2 levels showed no significant between-group differences at 12 months. C16 levels were significantly lower in responders from initiating of KDT to the end of study (C16, β = −0.70, *p* < 0.01 at baseline; β = −1.08, *p* < 0.005 at 3 months; β = −0.83, *p* < 0.001 at 9 months; β = −0.39, *p* < 0.05 at 12 months). C16:1 levels were significantly lower in responders at baseline, 3 months and 9 months (C16:1, β = −0.07, *p* < 0.05 at baseline; β = −0.07, *p* < 0.001 at 3 months; β = −0.05, *p* < 0.05 at 6 months). C18 levels were significantly lower in responders at baseline, 6 and 9 months (C18, β = −0.47, *p* < 0.001 at baseline; β = −0.51, *p* < 0.005 at 6 months; β = −0.29, *p* < 0.01 at 9 months). However, C18 levels showed no significant between-group differences at the endpoint of study. Our data disclosed that although blood levels of some long-chain acylcarnitines were significantly lower in responders at some time points of KDT; however, just palmitoyl carnitine (C16) levels were found significantly lower in responders at the endpoint of study.

### 3.13. Amino Acids in Responders

The comparison of amino acids between responders and non-responders was demonstrated in Table 3. Phenylalanine decreased in responders, reaching nadir levels at 6 months, and was significantly lower than in non-responders at 6 months and 9 months, but it tended to increase thereafter, although not to the baseline values by the end of the study (Phenylalanine, β = −11.97, *p* < 0.005 at 6 months; β = −3.81, *p* < 0.05 at 9 months). Levels of glycine increased significantly in responders from 3 months to 9 months, at which time they were significantly higher than the levels in non-responders (Glycine, β = 92.61, *p* < 0.001 at 3 months; β = 154.21, *p* < 0.01 at 6 months; β = 77.63, *p* < 0.05 at 9 months). Leucine/isoleucine levels in responders were significantly lower than that in non-responders at 3, 9, and 12 months of KDT (Leucine/isoleucine, β = −58.47, *p* < 0.001at 3 months; β = −37.16, *p* < 0.001 at 9 months; β = −36.19, *p* < 0.01 at 12 months). Phenylalanine/tyrosine ratio in responders was significantly lower than that in non-responders at 6 months and higher than that in non-responders at 12 months (phenylalanine/tyrosine ratio, β = −0.3, *p* < 0.001at 6 months; β = 0.21, *p* < 0.001 at 12 months). Arginine levels were significantly higher in responders at 6 months but not other time points (Arginine, β = 0.83, *p* < 0.05 at 6 months). Valine levels were significantly lower in responders at 9 months, but not at other time points (valine, β = −21.57, *p* < 0.05 at 9 months). Ornithine levels were significantly higher in responders at baseline (ornithine, β= 9.72, *p* < 0.05 at baseline), but it got paradoxically lower than that in non-responders at 3 months and 6 months (β = −7.56, *p* < 0.01 at 3 months; β = −5.79, *p* < 0.05 at 6 months). Amino acids, including tyrosine, proline, methionine, glutamic acid, citrulline, and homocysteine showed no significant between-group differences at different time points of KDT. Our results demonstrated that leucine/isoleucine levels, the ketogenic amino acids, in responders were significantly lower at the endpoint of study, which may indicate that degrading ketogenic amino acids into acetyl-CoA significantly decreased in responders at 12 months of KDT. Although phenylalanine/tyrosine ratio was significantly higher in responders at 12 months, the mean phenylalanine level in responders never reached ≥400 umol/l and the ratio never reached ≥1.5 at 12 months (mean phenylalanine level: 39.33 ± 6.85 umoL/L; mean ratio 1.14 ± 0.24). The relative high phenylalanine/tyrosine ratio in responders was presumptive no clinical significance.

### 3.14. Predictors and Associated Factors of KD Effectiveness

ROC analysis was applied to identify metabolic parameters that could predict the efficacy of KDT at baseline. The AUCs for C14, C3, and C18 were between 0.79–0.85 when comparing these metabolites at baseline between responders and non-responders (Figure 2A, C14 carnitine, AUC = 0.846, *p* < 0.01; C3 carnitine, AUC = 0.835, *p* < 0.005; C18 carnitine, AUC = 0.791, *p* < 0.01), indicating high predictive ability of KDT effectiveness. This result indicated that a lower baseline C14 level had the highest predictive ability of KD effectiveness.

After 12 months of KDT, there was a significant decrease in leucine/isoleucine, C16-and C5 carnitines at 12 months as shown in Table 2 and Table 3. The AUCs for leucine/isoleucine, C16-and C5 carnitines were between 0.7 and 0.9 when comparing responders and non-responders (Figure 2B, leucine/isoleucine, AUC = 0.856, *p* < 0.001; C16 carnitine, AUC = 0.800, *p* < 0.05; C5 carnitine, AUC = 0.756, *p* < 0.05), indicating strong association with KDT effectiveness at 12 months. Leucine-isoleucine had the highest AUC value, followed by C16 carnitine, and then C5 carnitine.

## 4. Discussion

Our results provide evidence that C2 carnitine level significantly increased throughout the diet therapy; otherwise, the long-chain acylcarnitines, C16 and C16:1 levels were significantly decreased after 12 months of KDT. Study subjects had significant lower carnitine levels than baseline values after 12 months of KDT, though carnitine level gradually elevated toward baseline level eventually. It is known that the brain utilizes ketone bodies, especially βHB, derived from acetyl-coenzyme A (acetyl-CoA) and acetoacetyl-coenzyme A (acetoacetyl-CoA) produced by β-oxidation of fatty acids in the mitochondria of the liver [25]. Long-chain fatty acids (>12 carbons) from KDT undergo thio-esterification to CoA and are catalyzed by acyl-CoA synthases (ACS) to form acyl-CoA [26]. Acyl-CoA must be conjugated to carnitine with formation of acyl-carnitines through the action of carnitine palmitoyl transferase I (CPTI), so that it can penetrate mitochondrial membrane. Acyl-carnitines are translocated across the inner mitochondrial membrane by acyl-carnitine translocase (CACT). Carnitine is removed from acyl-carnitine inside the mitochondrion by carnitine palmitoyl transferase II (CPTII), and acyl-CoAs and acetylcarnitine (C2 carnitine) are generated. Acyl-CoAs can enter β-oxidation in the mitochondrial matrix, finally producing acetyl-CoA, which leads to production of ketone bodies or to entry of citric acid cycle (Figure 3). The higher the efficiency of mitochondrial β-oxidation or energy metabolism in the liver, the more C2 carnitine is formed from free carnitine, leading to less acumination of long-chain fatty acid and a drop in plasma free carnitine. Thus, the decrement of C16 carnitine and the increment of C2 carnitine in our result may indicate improvement of mitochondrial functioning and energy metabolism after KD. The clinical practice with regard to carnitine supplement for a drop in free carnitine has huge diversity. We followed the recommendations given by KD expert consensus not routinely supplying carnitine for children with KDT unless they showed biochemical or symptomatic deficiency.

C3-, C5-, C5:OH-, and C4-DC carnitine, derives from ketogenic amino acids, were significantly lower than baseline levels, while C4:OH, derived from CoA-ester of the ketone bodies, was significantly higher than baseline values after 12 months of KDT. Since a ketogenic diet provides 80–90% of calories as lipid, ketogenesis is characterized by a decrease in a protein breakdown and an increase in fat and ketone use [27]. It can provide explanation of opposite alteration of ketogenic amino acids (e.g., C3-C5-C5:OH and C4-DC) and C4:OH, the derives of ketone body.

We disclosed C8 and C8/C10 ratio became significantly lower after 12 months of KDT. C8/C10 ratio is one of the most commonly reported markers in screening for medium-chain acyl-CoA dehydrogenase deficiency (MCADD) [28]. However, none of our study subjects conformed diagnosis of MCADD. Previous report revealed that C8 has the highest net ketogenic effect compared with other tested oils containing medium-chain triglyceride [23]. We presumed that C8 levels and C8/C10 ratio became lower after KDT may be due to rapidly turnover of C8 into ketone bodies. Both C14:1 and C14:2 carnitine levels trended upwards during study period. Elevated C14:1 and C14:2 plasma acylcarnitine after a controlled fast is a diagnostic strategy for very long-chain acyl-coA dehydrogenase (VLCAD) deficiency. However, the elevation of C14:1 and C14:2 plasma acylcarnitines in the setting of prolonged fast and hypoglycemia with induction of lipolysis and generating ketone response is physiologic and not indicative of a diagnosis of VLCAD deficiency [29,30]. Since ketogenic diet is a high fat and low carbohydrate diet which mimics starvation by generating ketone bodies, the elevation of C14:1 and C14:2 may reflect lipolysis getting progressively increasing as KDT keeping on going.

KDT affects not only lipids but also amino acids. There have been several reports on changes in CSF levels of amino acids under KDT [31,32]; however, literature is limited on changes in serum amino acid levels in response to KDT. Millichap found that the only change in serum under KDT was an increased leucine level [15,33]. Levels of several amino acids, including phenylalanine, tyrosine, leucine/isoleucine, and methionine, were also significantly lower after 12 months of KDT. Glycine is the only amino acid showing persistently high through study period in our study. Phenylalanine and tyrosine are the substrates of norepinephrine, which is considered one of the mechanisms of the anticonvulsant effect of KDT [34,35,36]. Consuming phenylalanine and tyrosine to form norepinephrine is presumed to be the possible explanation for their lower levels in our result, and this suggests that norepinephrine is one of the mechanisms of anticonvulsant effect. In addition, methionine is a substrate for biogenesis of carnitine, consumption of carnitine to form acyl-carnitine would lead to consumption of methionine, which explains the decrease in the methionine level [10,37]. Although there are reports of KDT being an effective treatment for refractory epilepsy in patients with non-ketotic hyperglycemia [38,39], we suggested that KDT should be avoided in these patients, given the persistently high glycine level based on our results.

We furthermore explored the association between metabolic parameters and the response to KDT. Schoeler et al. [9] reported that C2 carnitine at baseline was significantly higher in responders. However, in our data, responders had significantly lower C2 level at baseline, but its concentration became higher than non-responders after 12 months of KDT. The result was not consistent with the previous report, which demonstrated baseline C2 carnitine is positively associated with greater efficacy of KDT [9]. The most probable explanation for the contradictory results is presumed to be that patients in this study received non-fasting KDT, and the production of acetyl-carnitine to form ketones slowly progressed with time but not in a short time.

Additionally, our data disclosed that C3, C14, C18 at baseline, while C3, C5, C16, and leucine-isoleucine at 12 months are negatively associated with greater efficacy of KDT. Branch-chain amino acids (BACC), such as leucine isoleucine and valine are metabolic fuels that can generate C3- and C5 acylcarnitines. C5 acylcarnitines are comprised of α-methylbutyryl and isovalerylcarnitine species, intermediates in mitochondrial isoleucine and leucine catabolism. C3-acylcarnitine reflects the propionyl CoA pool, which is a byproduct of both isoleucine and valine catabolism in the mitochondria. It was reported that accumulation of C3-and C5-acylcarnitines could contribute to incomplete oxidation of fatty acids. Responders and non-responders were different in long-, C3 and C5 acylcarnitine indicating they may be distinct in the aspect of mitochondrial functioning. We have more non-responders receiving VPA during study period. The relatively higher long-chain fatty acid, C14, C16, C16:1, C18, C3, and C5 found in non-responders might result from the treatment with VPA. VPA is a fatty acid analog and a substrate for the fatty acid β-oxidation pathway. VPA combines with CoA or carnitine to form valproyl-CoA and then leads to direct competitive inhibition of carnitine uptake at the transporter site by reducing the availability of free CoA. Mitochondrial depletion of free CoA impairs β-oxidation of fatty acids and impairs subsequent transport of long-chain fatty acids [40]. Some studies have shown that VPA induces accumulation of long-chain fatty acids (C10–16) in cultured fibroblasts [41] or in humans (C12, C14:1, C16;1, C18:1) [42]. Our results suggested that accumulation of long-chain fatty acids, C3 and C5 acylcarnitine is an indicator of impaired fatty acid β-oxidation. Responders with lower level of long-chain fatty acid and C3 at initiation and long-term management of KDT are assumed to have a better mitochondrial β-oxidation pathway, which also acted as associated factors for greater effectiveness of KDT in our study.

Responders in this study demonstrated hypocarnitinemia throughout the study. Factors believed to increase the risk of hypocarnitinemia in patients with epilepsy include young age, use of multiple AEDs, and poor nutritional status [43,44,45]. Although our data indicated a positive correlation between body mass index (BMI) and the free carnitine level (Appendix A); however, the tendency of free carnitine and BMI was not similar in this study. Free carnitine level decreased gradually, reaching a nadir at 6 months of therapy, whereas BMI in responders gradually decreased between 9 and 12 months. This finding suggests that hypocarnitinemia in responders cannot be attributed to low BMI. Researchers have found carnitine deficiency in 4–76% of patients receiving VPA [46,47,48]. Although more non-responders than responders received VPA as monotherapy or add-on treatment in our study, hypocarnitinemia only occurred in responders. Therefore, treatment with VPA could not be a reason for hypocarnitinemia in patients on KDT in this study. There is another biochemical reason that can explain KDT-related hypocarnitinemia. We presumed that responders possessed a better functioning mitochondrial β-oxidation pathway, resulting in consuming plasma free carnitine, which provided the best explanation of hypocarnitinemia in responders.

Like previous researches on KDT, this study has some limitations, including a small sample size, lack of blinding or randomization, lack of a control group, and lack of age stratification. The human metabolome is influenced by a number of endogenous factors, such as age, sex, and BMI [49,50,51,52], so that age stratification is mandatory if we can extend the study numbers. Blinding is difficult in KDT research for obvious reasons. The effectiveness of KDT in patients with refractory epilepsy is well documented, so it would be ethically to randomize patients to study groups. Patients with refractory epilepsy account for only 30% of all cases of pediatric epilepsy, so it is inevitable that those receiving KDT are a small population. Finally, given the small sample size, we could not stratify our patients by age.

## 5. Conclusions

The novel finding of this study is that C3-, C5-, C5:OH-, and C4-DC decreased, and C4OH increased significantly at 12 months of KDT indicated decreasing glucose breakdown with increasing ketosis on long-term KDT. We also suggested that KDT should be avoided in patients with non-ketotic hyperglycemia, given the persistently high glycine level based on our results. Transient hypocarnitinemia was found during 9 months of KDT but was asymptomatic and resolved eventually, so that there is no need for carnitine supplementation. Our results also afford a new insight into the metabolic parameters that predict the response to KDT. The lower C3, C14, C18 at baseline could predict the efficacy of KDT, whereas lower C5, C16, and Ile/Leu level were factors associated with the effectiveness of long-term KDT. The results indicated the better mitochondrial βoxidation function is associated with greater KDT response.

## Figures and Tables

**Figure 1 nutrients-13-00021-f001:**
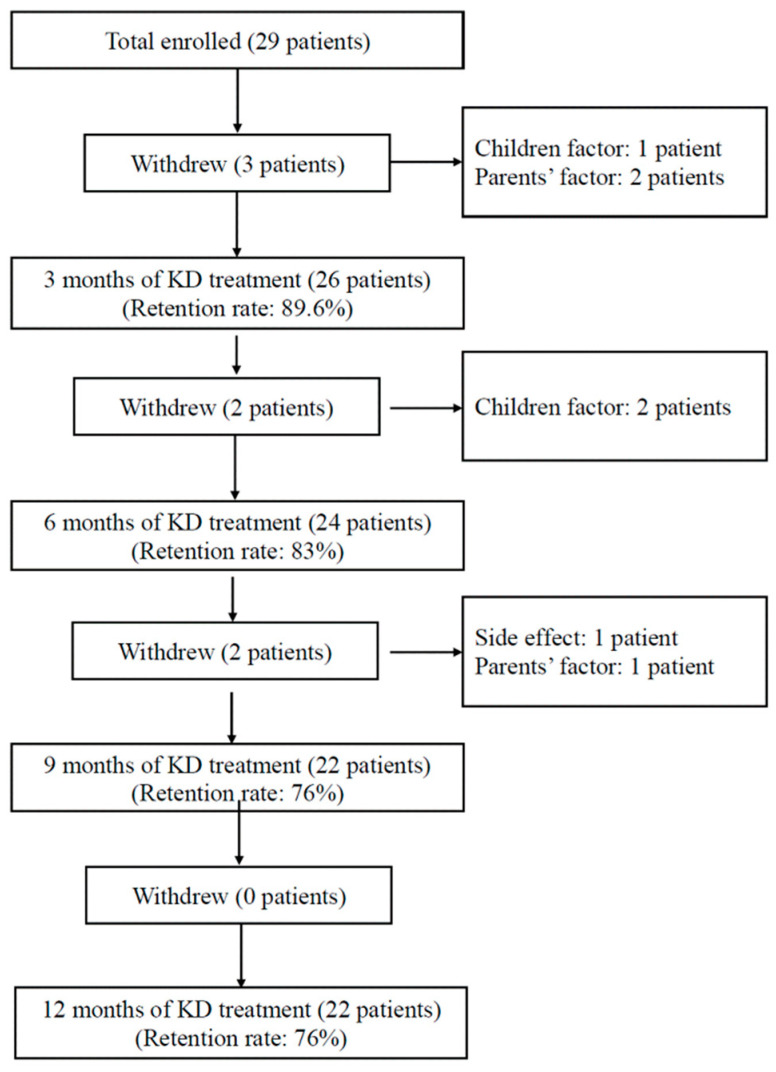
Flow chart showing the study protocol.

**Figure 2 nutrients-13-00021-f002:**
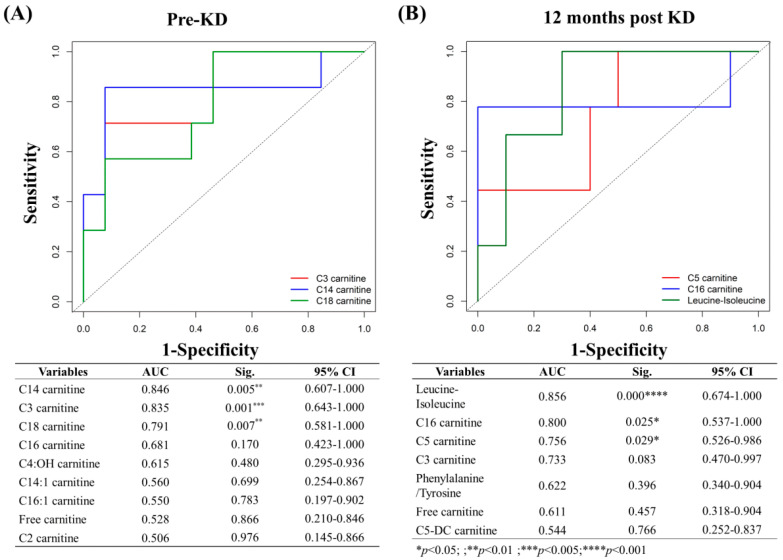
Receiver-operating characteristic (ROC) analysis of the discriminant metabolites between groups. (**A**) Pre-KD; (**B**) 12 months post KD. Individual metabolites of significance (solid lines) are shown along with the reference (dotted line). Area under the curve (AUC) values, significance level, and 95% confidence interval are shown. (** p* < 0.05; *** p* < 0.01; **** p* < 0.005; ***** p* < 0.001).

**Figure 3 nutrients-13-00021-f003:**
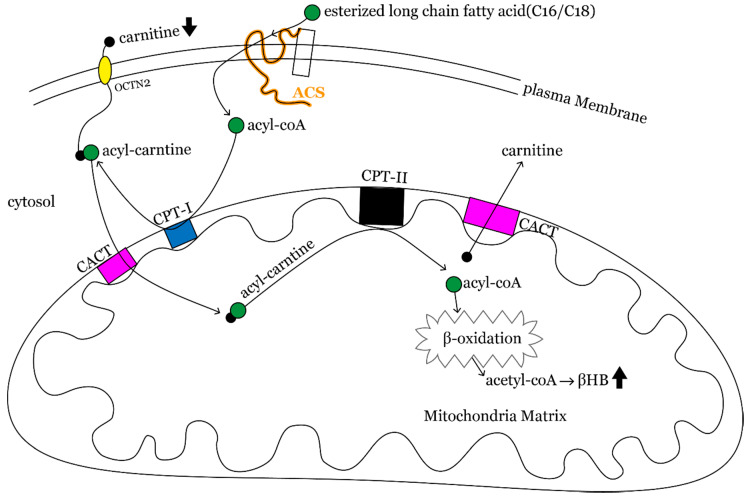
Fatty acid β-oxidation pathway. Fatty acids (especially long-chain fatty acids) undergo acylation, a process catalyzed by acyl-CoA synthases that traps them in the cytoplasm as acyl-CoA thioesters. Acyl-CoA thioesters then conjugate to carnitine with formation of acylcarnitines through the action of carnitine palmitoyl transferase I. Acylcarnitines are translocated across the inner mitochondrial membrane by acylcarnitine translocase. Carnitine can be removed from acylcarnitine by carnitine palmitoyl transferase II inside the mitochondria, and acyl-CoAs are re-generated. Acyl-CoAs can enter β-oxidation in the mitochondrial matrix with final products of acetyl-CoA that produce ketone bodies (mainly βHB) in the liver.

**Table 1 nutrients-13-00021-t001:** Fatty acid acylcarnitines and amino acids at different time point of KDT for 22 study subjects.Fatty acid acylcarnitines and amino acids were checked at baseline and after 3, 6, 9, and 12 month of ketogenic diet therapy. All statistically significant values are bolded.

**Fatty Acid, *n* = 22**																					
	**Free Carnitine**			**C2 Carnitine**			**C3 Carnitine**			**C4:OH Carnitine**			**C4-DC Carnitine**			**C5 Carnitine**			**C5:OH Carnitine**		
**Months**	**β**	**SE**	***p***	**β**	**SE**	***p***	**β**	**SE**	***p***	**β**	**SE**	***p***	**β**	**SE**	***p***	**β**	**SE**	***p***	**β**	**SE**	***p***
0	0.00	-	-	0.00	-	-	0.00	-	-	0.00	-	-	0.00	-	-	0.00	-	-	0.00	-	-
3	−3.08	2.07	0.136	7.29	2.40	0.002 ***	−0.64	0.24	0.006 **	0.25	0.04	0.000 ****	−0.25	0.04	0.000 ****	−0.02	0.01	0.253	−0.03	0.02	0.093
6	−3.91	2.06	0.057	8.15	2.80	0.004 ***	−0.80	0.19	0.000 ****	0.26	0.04	0.000 ****	−0.26	0.05	0.000 ****	−0.02	0.01	0.063	−0.05	0.02	0.005 **
9	−5.10	1.69	0.003 ***	5.12	2.46	0.037 *	−0.93	0.16	0.000 ****	0.23	0.04	0.000 ****	−0.30	0.04	0.000 ****	−0.02	0.01	0.044 *	−0.05	0.02	0.008 **
12	−4.07	1.97	0.039 *	7.14	3.12	0.022 *	−0.79	0.17	0.000 ****	0.24	0.05	0.000 ****	−0.31	0.04	0.000 ****	−0.02	0.01	0.022 *	−0.04	0.02	0.039 *
	**C8/C10 Carnitine**			**C14 Carnitine**			**C14:1 Carnitine**			**C14:2 Carnitine**			**C16 Carnitine**			**C16:1 Carnitine**					
**Months**	**β**	**SE**	***p***	**β**	**SE**	***p***	**β**	**SE**	***p***	**β**	**SE**	***p***	**β**	**SE**	***p***	**β**	**SE**	***p***			
0	0.00	-	-	0.00	-	-	0.00	-	-	0.00	-	-	0.00	-	-	0.00	-	-			
3	−0.05	0.07	0.531	0.02	0.03	0.521	0.03	0.01	0.005 ***	0.00	0.00	0.314	−0.12	0.13	0.333	−0.01	0.01	0.296			
6	−0.02	0.11	0.822	−0.03	0.01	0.002 ***	0.00	0.01	0.741	0.00	0.01	0.855	−0.26	0.10	0.009 **	−0.02	0.01	0.025 *			
9	−0.04	0.07	0.591	−0.02	0.01	0.040 *	0.02	0.01	0.070	0.00	0.00	0.896	−0.31	0.11	0.007 **	−0.02	0.01	0.083			
12	−0.15	0.05	0.004 ***	−0.02	0.01	0.151	0.03	0.02	0.113	0.01	0.01	0.035 *	−0.38	0.12	0.001 ***	−0.03	0.01	0.027 *			
**Amino Acid, *n*=22**																					
	**Phenylalanine**			**Tyrosine**			**Glycine**			**Proline**			**Leucine-Isoleucine**			**Methionine**			**Arginine**		
**Months**	**β**	**SE**	***p***	**β**	**SE**	***p***	**β**	**SE**	***p***	**β**	**SE**	***p***	**β**	**SE**	***p***	**β**	**SE**	***p***	**β**	**SE**	***p***
0	0.00	-	-	0.00	-	-	0.00	-	-	0.00	-	-	0.00	-	-	0.00	-	-	0.00	-	-
3	−6.61	2.00	0.001 ****	−12.15	3.29	0.000 ****	78.21	22.39	0.000 ****	−26.54	17.18	0.122	−35.61	19.59	0.069	−3.64	1.40	0.009 **	−0.27	0.46	0.552
6	−7.92	2.37	0.001 ****	−14.75	2.91	0.000 ****	91.01	37.92	0.016 *	−52.66	13.52	0.000 ****	−56.41	18.67	0.003 ***	−4.37	1.63	0.007 **	−0.90	0.46	0.048 *
9	−9.28	1.81	0.000 ****	−11.77	2.72	0.000 ****	74.71	23.49	0.001 ***	−49.16	14.48	0.001 ****	−52.51	13.75	0.000 ****	−5.28	1.48	0.000 ****	−0.55	0.45	0.221
12	−6.63	1.89	0.000 ****	−13.79	2.94	0.000 ****	49.82	17.93	0.005 **	−29.88	16.82	0.076	−48.35	14.31	0.001 ****	−5.48	1.19	0.000 ****	−0.53	0.49	0.278
	**Valine**			**Citrulline**			**Homocysteine**														
**Months**	**β**	**SE**	***p***	**β**	**SE**	***p***	**β**	**SE**	***p***												
0	0.00	-	-	0.00	-	-	0.00	-	-												
3	−23.39	21.37	0.274	2.57	1.26	0.041 *	1.17	0.52	0.025 *												
6	−37.94	22.61	0.093	−1.00	1.47	0.497	0.54	0.63	0.387												
9	−38.38	17.38	0.027 *	1.06	1.35	0.432	0.34	0.49	0.486												
12	−41.06	17.11	0.016 *	−0.51	1.50	0.732	−0.23	0.47	0.630												

β: regression weight; SE: Standard error; * *p* < 0.05, ** *p* < 0.01, *** *p* < 0.05, **** *p* < 0.001.

**Table 2 nutrients-13-00021-t002:** Comparison of fatty acid acylcarnitines between responders and non-responders. Acylcarnitines profile of responders (*n* = 12) and non-responders (*n* = 10). Group differences were analyzed by conducting generalized estimating equation (GEE) after adjusting confounding effect of anticonvulsants. *R*: responders; *NR*: non-responders.

	Time (Month)	β	SE	*p*		Time (Month)	β	SE	*p*
BHB	0	0.37	0.87	0.673	C6-DC carnitine	0	0.01	0.01	0.102
	3	0.31	0.57	0.588		3	0.00	0.00	0.324
	6	1.43	0.64	0.025 *		6	−0.01	0.01	0.031 *
	9	−0.65	0.74	0.384		9	−0.01	0.00	0.275
	12	−0.06	0.77	0.934		12	0.00	0.00	0.465
Free carnitine	0	−7.85	3.73	0.035 *	C8 carnitine	0	−0.01	0.03	0.657
	3	−11.84	4.05	0.003 ***		3	−0.07	0.07	0.333
	6	−15.94	4.18	0.000 ****		6	0.01	0.01	0.405
	9	−6.77	3.58	0.059		9	−0.06	0.01	0.000 ****
	12	−8.59	3.11	0.006 **		12	−0.01	0.01	0.230
C2 carnitine	0	−8.95	3.29	0.006 **	C10 carnitine	0	−0.02	0.04	0.688
	3	−9.67	3.84	0.012 *		3	−0.06	0.05	0.240
	6	−2.65	4.50	0.556		6	0.02	0.02	0.333
	9	−1.60	4.96	0.747		9	−0.07	0.02	0.000 ****
	12	1.464	5.3079	0.783		12	−0.03	0.02	0.112
C3 carnitine	0	−1.28	0.60	0.032 *	C10:1 carnitine	0	0.01	0.03	0.664
	3	−1.68	0.34	0.000 ****		3	0.02	0.02	0.355
	6	−1.80	0.30	0.000 ****		6	0.02	0.02	0.144
	9	−1.12	0.51	0.027 *		9	−0.08	0.01	0.000 ****
	12	−0.98	0.43	0.022 *		12	−0.02	0.02	0.253
C4 carnitine	0	−0.05	0.06	0.456	C8/C10 carnitine	0	−0.04	0.17	0.817
	3	−0.21	0.06	0.000 ****		3	0.07	0.15	0.628
	6	−0.14	0.06	0.028 *		6	−0.02	0.11	0.846
	9	−0.10	0.06	0.084		9	0.07	0.13	0.575
	12	−0.03	0.05	0.547		12	−0.01	0.12	0.949
C4:OH carnitine	0	−0.07	0.03	0.033 *	C12 carnitine	0	−0.02	0.02	0.337
	3	−0.04	0.08	0.617		3	−0.07	0.03	0.027 *
	6	0.06	0.07	0.404		6	−0.02	0.01	0.151
	9	−0.07	0.08	0.379		9	−0.03	0.02	0.055
	12	0.09	0.11	0.409		12	−0.01	0.02	0.560
C4-DC carnitine	0	−0.15	0.09	0.108	C14 carnitine	0	−0.04	0.02	0.022 *
	3	−0.16	0.08	0.043 *		3	−0.11	0.08	0.146
	6	−0.06	0.10	0.500		6	−0.04	0.01	0.007 **
	9	0.05	0.08	0.550		9	−0.04	0.02	0.016 *
	12	0.04	0.11	0.712		12	0.00	0.01	0.979
C5 carnitine	0	−0.02	0.04	0.592	C14:1 carnitine	0	−0.04	0.01	0.000 ****
	3	−0.09	0.02	0.000 ****		3	−0.05	0.02	0.022 *
	6	−0.10	0.01	0.000 ****		6	0.00	0.02	0.911
	9	−0.07	0.02	0.002 ***		9	−0.05	0.02	0.009 **
	12	−0.04	0.01	0.002 ***		12	−0.01	0.02	0.564
C5:1 carnitine	0	0.00	0.00	0.275	C14:2 carnitine	0	0.00	0.01	0.896
	3	−0.01	0.00	0.013 *		3	−0.01	0.01	0.153
	6	0.00	0.00	0.401		6	0.00	0.01	0.831
	9	−0.01	0.00	0.001 ***		9	−0.03	0.01	0.003 ***
	12	0.00	0.00	0.504		12	0.00	0.01	0.761
C5:OH carnitine	0	−0.08	0.09	0.371	C16 carnitine	0	−0.70	0.25	0.005 **
	3	−0.12	0.04	0.006 **		3	−1.08	0.35	0.002 ***
	6	−0.10	0.05	0.062		6	−0.83	0.23	0.000 ****
	9	−0.11	0.04	0.012 *		9	−0.62	0.21	0.003 ***
	12	−0.11	0.09	0.224		12	−0.39	0.16	0.013 *
C5-DC carnitine	0	0.01	0.01	0.055	C16:1 carnitine	0	−0.07	0.03	0.011 *
	3	−0.01	0.00	0.003 ***		3	−0.07	0.01	0.000 ****
	6	0.00	0.00	0.232		6	−0.05	0.02	0.014 *
	9	−0.02	0.01	0.011 *		9	−0.05	0.03	0.059
	12	−0.01	0.00	0.005 ***		12	−0.01	0.01	0.235
C6 carnitine	0	−0.01	0.02	0.620	C18 carnitine	0	−0.47	0.08	0.000 ****
	3	−0.09	0.06	0.152		3	−0.50	0.30	0.093
	6	−0.02	0.00	0.000 ****		6	−0.51	0.18	0.004 ***
	9	−0.02	0.01	0.031 *		9	−0.29	0.11	0.006 **
	12	0.00	0.01	0.836		12	−0.22	0.16	0.168

β: regression weight; SE: Standard error; * *p* < 0.05, ** *p* < 0.01, *** *p* < 0.05, **** *p* < 0.001.

**Table 3 nutrients-13-00021-t003:** Comparison of amino acids between responders and non-responders. Amino acids profile of responders (*n* = 12) and non-responders (*n* = 10). Group differences were analyzed by GEE after adjusting confounding effect of anticonvulsants. *R*: responders; *NR*: non-responders.

	Time (month)	β	SE	*p*		Time (month)	β	SE	*p*
Phenylalanine	0	0.18	3.99	0.964	Arginine	0	1.52	1.06	0.154
	3	−3.35	1.78	0.060		3	0.63	1.08	0.560
	6	−11.94	3.74	0.001 ***		6	0.83	0.32	0.010 **
	9	−3.81	1.63	0.019 *		9	0.83	0.76	0.272
	12	0.21	3.00	0.943		12	0.74	0.46	0.107
Tyrosine	0	4.84	7.65	0.527	Phenylalanine	0	−0.04	0.09	0.618
	3	−4.09	5.68	0.472	/Tyrosine	3	−0.06	0.18	0.719
	6	−0.99	2.38	0.677		6	−0.30	0.06	0.000 ****
	9	1.38	4.20	0.743		9	−0.12	0.08	0.144
	12	−7.21	3.77	0.056		12	0.21	0.06	0.000 ****
Glycine	0	38.49	25.29	0.128	Alanine	0	−27.17	44.74	0.544
	3	92.61	25.09	0.000 ****		3	−13.96	38.93	0.720
	6	154.21	58.87	0.009 **		6	−13.97	32.42	0.666
	9	77.63	38.82	0.046*		9	15.19	21.28	0.475
	12	53.38	33.84	0.115		12	28.74	25.08	0.252
Proline	0	−1.84	33.52	0.956	Valine	0	−8.54	29.85	0.775
	3	−0.04	31.20	0.999		3	−23.79	20.00	0.234
	6	−7.89	21.46	0.713		6	−12.92	15.43	0.402
	9	−23.27	15.74	0.139		9	−21.57	10.83	0.046 *
	12	−37.73	30.29	0.213		12	−21.74	18.87	0.249
Leucine	0	−9.71	22.42	0.665	Ornithine	0	9.72	4.70	0.039 *
-Isoleucine	3	−58.47	8.09	0.000 ****		3	−7.56	2.82	0.007 **
	6	−15.30	10.54	0.147		6	−5.79	2.71	0.033 *
	9	−37.16	8.81	0.000 ****		9	−6.91	9.64	0.473
	12	−36.19	13.60	0.008 **		12	3.15	4.83	0.515
Methionine	0	−2.34	3.26	0.473	Citrulline	0	−3.28	3.06	0.284
	3	−4.81	2.82	0.088		3	−3.58	2.18	0.101
	6	−1.38	2.90	0.635		6	−2.28	2.35	0.332
	9	0.57	1.19	0.629		9	−1.91	3.02	0.528
	12	−2.49	1.64	0.129		12	1.17	2.26	0.603
Glutamic acid	0	−3.61	12.51	0.773	Homocysteine	0	−0.03	0.48	0.946
	3	−20.97	14.72	0.154		3	−0.44	0.94	0.640
	6	5.27	10.47	0.615		6	0.55	1.18	0.643
	9	6.36	11.81	0.590		9	−1.42	0.64	0.027 *
	12	17.12	10.88	0.116		12	−0.69	1.09	0.524

β: regression weight; SE: Standard error; * *p* < 0.05, ** *p* < 0.01, *** *p* < 0.05, **** *p* < 0.001.

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
