# Peer review of "An Examination of Serum Acylcarnitine and Amino Acid Profiles at Different Time Point of Ketogenic Diet Therapy and Their Association of Ketogenic Diet Effectiveness"

_nutrients, 2020, doi:10.3390/nu13010021_

Round 1

Reviewer 1 Report

The manuscript contains detailed  information concerning metabolism of acylcarnitines and aminoacids during a kind of ketogenic dietary therapy .  This date are very useful as there are contrasting opinion on the need for carnitine supplementation during long-term ketogenic therapies. However the kind of diet used must be better described.  The predictive values of C3,C14,C18 at baseline is interesting but has a lower clinical value because few labs can effectively perform the measurements.

Title The title is misleading because the reader expect to find a full spectrum of biochemical blood analysis but the report concerns specifically plasma acylcarnitines and amino acids and it is better to specify

Abstract  The term KDT refer to any dietary regimen inducing ketosis, please specify which KDT has been used: classic KD (cKD) MCT KT, MAD or else?  There are acronyms first used in the abstract that are not explained ( C3-, C5-, C5:OH-, C4-DC,  NHK )

Introduction The term KDT refer to any dietary regimen inducing ketosis, please specify which KDT has been used: classic KD (cKD) MCT KT, MAD or else?

Methodology The ketogenic ratio indicated in the methodology section is representative of a classic KD but the amount of protein is rather high compared to the usual composition.  Please specify the percentage of CHO, lipids and protein. The amount of protein clearly influences the circulating amino-acids. Also the kind of protein food selected in the dietary regimen ( animal or vegetal proteins) needs to be specified

Table 1 Too many figures the tables need to be simplified

Author Response

Dear reviewer, please see the attached file which is a point-by-point response to the reviewer's comments. 

Reviewer 2 Report

The aim of this study was to identify predictors and factors associated with the effectiveness of ketogenic diet therapy in drug resistant epilepsy. The finding were: some short-chain carnitines has persistently lower levels in responders than in non-responders after KDT. Medium-chain carnitines did not show differences between responders and non-responders at endpoint of study. Some long-chain carnitines were lower in responders at some time points.

Some issues that could be commented on more were:

  1. There was a very large age range from infants to 19 years old. Was KDT administered as formula for some subjects and how could this affect the results?
  2. In Table 1, beta is not defined, units are not provided, and acronyms for the anticonvulsants are not provided.
  3. It is not clear what is being shown in Figure 2 and how sensitivity and specificity were calculated. The dotted line is not visible.
  4. The supplementary data shows increased seizures with KDT in a few subjects.
  5. It is mentioned that there is not enough data to stratify based on age, but could more discussion be added regarding how expected age- and sex-dependent changes in carnitine levels could confound the results.
  6. Spell and grammar check are required. A few examples:

Line 176 “on the other hand” not “in the other hand”

Line 177 “explain how” not “explain”

Line 183 “turned over” not “turnover”

Line 186 “not” instead of “no”

Line 194 during “the” study period

Etc.

Author Response

Reviewer 2

The aim of this study was to identify predictors and factors associated with the effectiveness of ketogenic diet therapy in drug resistant epilepsy. The finding were: some short-chain carnitines has persistently lower levels in responders than in non-responders after KDT. Medium-chain carnitines did not show differences between responders and non-responders at endpoint of study. Some long-chain carnitines were lower in responders at some time points.

Some issues that could be commented on more were:

Point 1: There was a very large age range from infants to 19 years old. Was KDT administered as formula for some subjects and how could this affect the results?

Respond 1: Yes, formula made by dietitian was provided for those younger than 2 years old or patients receiving tube feeding. However, the keto ratio and energy amount of formula still followed the GRAD-KD protocol in this study, so that the results are not affected.  

Point 2. In Table 1, beta is not defined, units are not provided, and acronyms for the anticonvulsants are not provided.

Respond 2: We defined the beta as “ regression weight” under Table 1, and since beta is regression weight, it is lack of unit. In addition, we also defined SE and P value under Table 1. All abbreviations for the anticonvulsants were all deleted, since they never presented in Table 1.Please see the change on page 6.

Point 3. It is not clear what is being shown in Figure 2 and how sensitivity and specificity were calculated. The dotted line is not visible.

Respond 3: I am sorry for missing the dotted line, and I added the dotted line in the new Figure 2, please see page 11. The term ROC stands for Receiver Operating Characteristic, and it is a useful tool in assessing the diagnostic accuracy of any variable with a continuous spectrum of results. We assessed the predictive accuracy for the true (responders)/false (non-responders) situations in this study. The area under an ROC curve (AUC or AUROC) is a measure of the usefulness of a test, where a greater area means a more useful test. AUROC can also be used as a criterion to measure the test’s discriminative ability just like our study did. We found the best cuff-off point by Youden Index, and then we calculated the sensitivity and specificity based on the best cuff-off point. The statistical analysis was undertaken by R software (version 3.5.2; R Project for Statistical Computing, https://www.r-project.org).

Point 4. The supplementary data shows increased seizures with KDT in a few subjects.

Respond 4. Yes. Case 18 is a female baby born with severe hypoxic-ischemic encephalopathy. Brain MRI of this baby demonstrated diffusely brain parenchymal damage. Her initial seizure was focal motor seizure, and then evolved into multiple seizure types. She is really a patient with refractory epilepsy which is pharmacoresistant as well as ketogenic diet –resistant.

Point 5. It is mentioned that there is not enough data to stratify based on age, but could more discussion be added regarding how expected age- and sex-dependent changes in carnitine levels could confound the results.

Respond 5: I added four references (reference 51-54) regarding how metabolomics profiles are affected by age, sex and BMI in the text. Please see page 15, line 476-478.

Point 6. Spell and grammar check are required. A few examples:

Line 176 “on the other hand” not “in the other hand”

Line 177 “explain how” not “explain”

Line 183 “turned over” not “turnover”

Line 186 “not” instead of “no”

Line 194 during “the” study period

Etc.

Respond 6: The spelling and grammar were checked by Microsoft Word, and all the spell and grammar errors were corrected in red color as you can see in the text. We also corrected the spell and grammar errors as you pointed it out.

Line 182 (not 176) “in the other hand” had been corrected to “on the other hand”

Line 183 (not 177) “explain” had been corrected to “explain how”

Line 189 (not 183) “turnover” had been corrected to “turned over”

Line 192 (not 186) I wrote “not” instead of “no”

Line 200 (not 194) I corrected the words in the end of the sentence as “during the study period”

Round 2

Reviewer 2 Report

The authors have addressed the review concerns.